# Genomic Landscape and Mutational Spectrum of ADAMTS Family Genes in Mendelian Disorders Based on Gene Evidence Review for Variant Interpretation

**DOI:** 10.3390/biom10030449

**Published:** 2020-03-13

**Authors:** John Hoon Rim, Yo Jun Choi, Heon Yung Gee

**Affiliations:** 1Department of Pharmacology, Brain Korea 21 PLUS Project for Medical Sciences, Yonsei University College of Medicine, Seoul 03722, Korea; johnhoon1@yuhs.ac (J.H.R.) yjchoi1917@yuhs.ac (Y.J.C.); 2Department of Medicine, Physician-Scientist Program, Yonsei University Graduate School of Medicine, Seoul 03722, Korea

**Keywords:** ADAMTS genes, Mendelian disorders, genomic landscape

## Abstract

ADAMTS (a disintegrin and metalloproteinase with thrombospondin motifs) are a family of multidomain extracellular protease enzymes with 19 members. A growing number of ADAMTS family gene variants have been identified in patients with various hereditary diseases. To understand the genomic landscape and mutational spectrum of ADAMTS family genes, we evaluated all reported variants in the ClinVar database and Human Gene Mutation Database (HGMD), as well as recent literature on Mendelian hereditary disorders associated with ADAMTS family genes. Among 1089 variants in 14 genes reported in public databases, 307 variants previously suggested for pathogenicity in Mendelian diseases were comprehensively re-evaluated using the American College of Medical Genetics and Genomics (ACMG) 2015 guideline. A total of eight autosomal recessive genes were annotated as being strongly associated with specific Mendelian diseases, including two recently discovered genes (*ADAMTS9* and *ADAMTS19*) for their causality in congenital diseases (nephronophthisis-related ciliopathy and nonsyndromic heart valve disease, respectively). Clinical symptoms and affected organs were extremely heterogeneous among hereditary diseases caused by ADAMTS family genes, indicating phenotypic heterogeneity despite their structural and functional similarity. *ADAMTS6* was suggested as presenting undiscovered pathogenic mutations responsible for novel Mendelian disorders. Our study is the first to highlight the genomic landscape of ADAMTS family genes, providing an appropriate genetic approach for clinical use.

## 1. Introduction

ADAMTS proteases, a superfamily of 19 secreted molecules, are zinc metalloendopeptidases; most ADAMTS protease substrates are extracellular matrix components including procollagen, von Willebrand factor, aggrecan, versican, brevican, and neurocan [1]. Since ADAMTS genes share similar structure and catalytic activity, ADAMTS proteins are known to participate in common biological processes, such as skin and cardiac development, connective tissue maintenance, and hemostasis [2]. While various perspectives of ADAMTS family genes have been studied for their clinical significance [3,4,5], a clinical genetic study focusing on their causality in Mendelian disorders is lacking [6].

As high throughput genomic technologies have revolutionized clinical practices, the scope of medical decision making has become broader, with the promise of personalized precision healthcare based on individual genotypes [7]. Genes that were uninvestigated before the next-generation sequencing era appear to possess numerous variants in clinical cases when compared to reference sequences from population databases. Neither underestimation nor overestimation of gene-disease association is helpful for clinical practices or even basic genomic research [8]. Thus, a systematic evaluation with a comprehensive review of up-to-date evidence is essential for accurate and appropriate genetic understanding to enhance the clinical usefulness of genomic studies.

With the help of technological advancements in ADAMTS studies, growing evidence for the association of ADAMTS genes with unique Mendelian disorders raises the need for an accurate and comprehensive evaluation, especially from the genetic perspective. In this study, we evaluate all reported variants in the ClinVar and HGMD mutation databases, as well as recent literature on Mendelian disorders associated with ADAMTS family genes. The information provided in this study demonstrates that the clinical interpretation of all reported mutations in ADAMTS genes requires careful professional curation and application of the latest information to suggest the possibility of ADAMTS gene involvement in novel Mendelian disorders.

## 2. Materials and Methods

### 2.1. Collection of Reported ADAMTS Family Gene Variants

All reported ADAMTS family gene variants suggested as being involved in disease were collected from publicly available mutation databases, including the Human Gene Mutation Database (HGMD version 2019.4) [9] and ClinVar (accessed 2019 Dec) [10] (Figure 1). For the HGMD database, variants annotated as “DM” and “DM?” were included in the study since HGMD curators presumably asserted the pathogenicity of these variants. For ClinVar, variants annotated as “pathogenic” or “likely pathogenic”, including all small nucleotide variants and copy number variants, were compiled. The annotation and nomenclature of the variants were confirmed using the Mutalyzer Name Checker tool [11] based on clinically relevant transcripts with the longest transcript and exons in each gene.

### 2.2. Systematic Literature Review for ADAMTS Family Genes on Mendelian Disorders

A systematic online search was performed for publications using MEDLINE (PubMed) (www.ncbi.nlm.nih.gov/pubmed), Embase (www.embase.com), and the Cochrane Database of Systematic Reviews electronic databases (www.cochranelibrary.com). The search was performed from database inception until 31 December 2019. Data for clinical and genomic information, as well as functional studies on variants, were extracted from eligible publications. Two independent authors (J.H.R. and Y.J.C.) assessed articles by title, abstract, and full text. A total of 68 studies that fulfilled the selection criteria of (1) human mutation study (not mouse), (2) germline mutation study with single gene–disease relationship (not association study), and (3) DNA mutation study (not epigenetic or proteomic) were included for further evaluation.

### 2.3. Evaluating ADAMTS Family Gene Disease Associations

Evidence associating each gene with a specific Mendelian disorder was systematically assessed based on a comprehensive analysis of various aspects, including gene ontology, protein functional domain for mutation location, and expression patterns. The Online Mendelian Inheritance in Man (OMIM) database [12] was used to confirm the currently validated disease associations and inheritance patterns in ADAMTS family genes. The Human Phenotype Ontology (HPO) database [13] was utilized in the interpretation process for organ-specific symptoms of suggested Mendelian disorders. Gene ontology (GO) [14] for ADAMTS family genes was searched using the Database for Annotation, Visualization, and Integrated Discovery (DAVID) [15] and Ensembl Genome Browser [16]. Protein domains and expression patterns among various organs across ADAMTS family genes were compared using UniProt [17] and the Human Protein Atlas (www.proteinatlas.org) [18], respectively.

### 2.4. Pathogenicity Interpretation for Variant and Gene Evaluation

To evaluate the pathogenic potential of presumably pathogenic ADAMTS family gene variants, we applied the 2015 ACMG guideline for variant classification [19]. To enhance the analysis accuracy, we profiled all aspects of variants, including population allele frequency, prediction algorithm results, and conservation status across species. For gene evaluation in terms of cause-and-effect relationships for hereditary disorders, we applied two recently published prediction algorithms for missense variant burden (i.e., PER (pathogenic mutation enriched regions) viewer [20] and MTR (missense tolerance regions) viewer [21]) in ADAMTS genes. These tools provided a statistical framework to identify gene regions with missense variation intolerance or pathogenic mutation enriched regions. Additionally, oe values (the ratio of observed/expected number of loss-of-function variants in the specific gene) and pLI scores (probability of being loss-of-function intolerant as metrics to measure a transcript’s intolerance to variation) provided by the gnomAD database [22] were incorporated to broaden genomic understanding.

## 3. Results

### 3.1. Gene-Disease Association of ADAMTS Family Genes Based on Pathogenic Mutations

In order to identify definitive gene–disease associations among mutated ADAMTS family genes in Mendelian disorders, we reviewed all publications archived from mutation databases and systematic literature review. A total of eight ADAMTS family genes were revealed to have strong causality for various Mendelian disorders if mutated (Table 1). Although several recently updated reviews have already briefly covered six ADAMTS genes [5], our evidence suggested two additional ADAMTS genes with a high probability of involvement in different diseases [23,24]. Except for *ADAMTS3* and *ADAMTS10*, six genes presented a unique association with specific disease phenotypes, indicating low genetic heterogeneity for unique Mendelian disorders caused by corresponding ADAMTS gene mutations. Regarding inheritance modes, all ADAMTS mutations acted recessively due to the nature of enzymes since enzymes are mostly haplo-sufficient to alleviate heterozygote loss of function mutation. This phenomenon could also be related to the dominance of nonsense, frameshift, and splice-site mutations in terms of the mutational spectrum in most ADAMTS genes. While *ADAMTS13* possessed more than 200 pathogenic mutations with clinical validations, mutations in seven other ADAMTS genes were rarely reported, suggesting a very low prevalence of these ultra-rare diseases (Table 1).

When all clinical cases were reviewed for organ-specific phenotypes, HPO analysis revealed phenotypic heterogeneity among ADAMTS family genes (Figure 2). While ocular symptoms were the most common, a broad spectrum of eye diseases were caused by *ADAMTS10*, *ADAMTS17*, and *ADAMTS18* [25,26,27,28]. Since *ADAMTS10* and *ADAMTS17* consisted of common domains, including PLAC, clinical presentations in Weill–Marchesani syndrome caused by *ADAMTS10* mutations and Weill–Marchesani-like syndrome caused by *ADAMTS17* mutations shared similar ocular and skeletal phenotypes [26]. In contrast, phenotypes in Ehlers–Danlos syndrome and Hennekam syndrome patients completely differed and involved different organs, although *ADAMTS2* and *ADAMTS3* possess the same procollagen N-propeptidase domains [29,30]. Furthermore, *ADAMTS9* with a unique GON domain presented neural, hearing, and renal phenotypes if mutated, all of which are specifically caused by ciliary dysfunction [23].

### 3.2. Updated ADAMTS Family Genes Responsible for Mendelian Disorders; ADAMTS9 and ADAMTS19

Our group recently found *ADAMTS9* mutations are a cause of nephronophthisis-related ciliopathies (NPHP–RCs) [23]. NPHP–RCs are a group of inherited diseases associated with defects in primary cilium structure and function. In this study, homozygosity mapping and whole-exome sequencing identified two cases with homozygous mutations in *ADAMTS9*. A novel homozygous frameshift truncating mutation (c.4575_4576del; p.Gln1525Hisfs*60) in *ADAMTS9* was identified in an European patient with NPHP and early-onset end-stage renal disease (ESRD) with the Joubert syndrome phenotype, including symptoms of vermis aplasia and corpus callosum hypoplasia. In addition, the patient presented with proteinuria, deafness, atrial septal defects, coloboma, and short stature. An Arabic child from consanguineous parents also had ESRD from NPHP, resulting from an amino acid substitution (c.194C>G; p.Thr65Arg) at Thr65, a highly conserved residue for protein function. The patient exhibited proteinuria, deafness, hepatosplenomegaly, anemia, thrombocytopenia, short stature, and osteopenia. While ADAMTS9 is known to be a secreted extracellular metalloproteinase, an in vitro study showed that ADAMTS9-loss resulted in shortened cilia and defective sonic hedgehog signaling. Knockdown of *adamts9* in zebrafish recapitulated NPHP–RC phenotypes, including renal cysts and hydrocephalus. These findings suggest that the identified mutations in *ADAMTS9* cause NPHP–RC and that ADAMTS9 is required for the formation and function of primary cilia.

Another study on *ADAMTS19* recently identified pathogenic mutations as the cause of non-syndromic heart valve disease [24]. Exome sequencing of four affected individuals in two consanguineous families showed novel homozygous truncating mutations in *ADAMTS19* (homozygous deletion involving exons 1 to 8 and homozygous nonsense mutation (c.1984C > T; p.Arg662*)). The authors suggested that a unique feature results from these *ADAMTS19* mutations, causing only heart valve disorders without affecting any other organs; this indicates non-syndromic features.

### 3.3. Evidence-Based Evaluation for Gene-Disease Relationships in ADAMTS Genes

Various aspects of gene evaluation, including GO, protein domains, expression databases, and functional assay availability, were reviewed (Table 2). When commonly shared biological process and molecular function GO terms were evaluated in all ADAMTS family genes, diverse GO terms appeared across various groups of ADAMTS genes. While “proteolysis” and “collagen processes” were dominantly annotated for biological processes, “metalloendopeptidase”, “metal ion binding”, and “zinc ion binding” were reported in more than half of the ADAMTS family genes. As phenotypic variability was characteristic in the affected organ spectrum, expression patterns of ADAMTS family genes were heterogeneous. However, in vitro functional assays were only available for seven genes despite the commonly shared functions among ADAMTS family genes.

### 3.4. Mutational Spectrum of ADAMTS Family Genes in Current Databases

When mutation databases (HGMD and ClinVar) were searched for ADAMTS gene variants, a total of 286 SNVs and 21 CNVs were collected. While eight genes with a strong causality in specific Mendelian disorders presented 34 SNVs on average, other ADAMTS genes with currently deficient clinical evidence possessed up to four SNVs (Table 3). Percentages of frameshift/nonsense/splicing variants were dominant in most causative ADAMTS genes, although missense variants were more predominant in *ADAMTS18*.

To predict the possibility of a missense mutational burden at “hot spots” on ADAMTS genes, two recently validated prediction tools (i.e., PER-viewer [20] and MTR-viewer [21]) were utilized. As PER-viewer considers the positional conservation status among paralogs, each ADAMTS gene was compared to other genes within the ADAMTS family. Four families were incorporated using PER-viewer, and this algorithm suggested that there were no definitive pathogenic mutation enriched regions for missense mutations in any ADAMTS gene, according to the suggested threshold of Bonferroni-adjusted *p*-value below 0.05. In contrast, *ADAMTS6*, *ADAMTS9*, *ADAMTS10*, and *ADAMTS14* appeared to have missense mutation enriched regions according to MTR-viewer based on the criterion of MTR score under 0.6 and FDR-adjusted *p*-value under 0.1, as defined by algorithm developers. As missense mutations in *ADAMTS9* and *ADAMTS10* are validated for their pathogenicity in association with Mendelian disorders, *ADAMTS6* and *ADAMTS14* are expected to be responsible for currently undiscovered Mendelian disorders with pathogenic missense mutations. Furthermore, oe value for missense variants in *ADAMTS6* was relatively low among ADAMTS family genes, indicating the possibility of missense mutations to be elucidated in the future. Another oe value for loss of function (LoF) mutations provided by the gnomAD database, along with the pLI score from a previous version of gnomAD, suggested *ADAMTS2* and *ADAMTS6* as LoF genes if pathogenic mutations caused Mendelian disorders. As all reported *ADAMTS2* and *ADAMTS6* variants did not satisfy the pathogenicity grades, pathogenic LoF mutations with distinctive dysfunctions might cause severe Mendelian disorders or even embryo lethality.

### 3.5. Reassessment of Previously Reported Pathogenic Mutations in ADAMTS Genes

The ACMG variant interpretation guideline [19] was applied for previously reported pathogenic mutations (Table 4). The mutational spectrum of *ADAMTS13* pathogenic mutations, which was well-reviewed in recently published literatures, revealed a heterogeneous pattern across mutation positions and variant types (Appendix A). Among the diverse evidence in the ACMG guideline, allele frequency in the control database highlighted as predicting the prevalence of Mendelian disorders were ultra-rare. A total of 45 mutations were validated for their suggested pathogenicity (Table 4), while another 12 presumably annotated pathogenic variants provided insufficient evidence for pathogenicity interpretation (Appendix A). Positional distribution of pathogenic and likely pathogenic mutations were dispersed among seven genes (Figure 3).

## 4. Discussion

In this study, we identified a subset of ADAMTS family genes with strong evidence of being causative genes for specific Mendelian disorders. As a growing number of variants are detected by up-to-date sequencing technologies [7], ADAMTS family genes are expected to be highlighted for their association with human disorders, especially hereditary diseases [3,4,5]. Understanding the genomic landscape and mutational spectrum of ADAMTS family genes in clinical cases will not only benefit further characterization of ADAMTS family proteins in biology but the future development and application of therapeutics for Mendelian disorders caused by ADAMTS gene mutations.

A total of eight ADAMTS family genes are currently known to be responsible for Mendelian disorders. Since all ADAMTS proteins are metalloproteases, the autosomal recessive modes of all genes are compatible with the characteristics of ADAMTS enzymes [1]. Furthermore, the mutational spectrum of nonsense and frameshift mutations could be explained by the LoF mechanism in most genes, as expected. While *ADAMTS13* mutations are well studied for heterogenous mutations across the gene in more than 200 congenital thrombotic thrombocytopenic purpura patients [31,32], only small numbers of patients and mutations have been reported in the other seven genes. Although the prevalence of ultra-rare Mendelian diseases caused by ADAMTS genes is expected [5], it is important to broaden the genetic spectrum of these genes as more patients with pathogenic ADAMTS gene mutations will present in the future. It is noteworthy that two recently discovered Mendelian disorders arising from *ADAMTS9* and *ADAMTS19* broadened the spectrum of affected organs (renal and cardiac diseases, respectively) that have never been associated with six previously identified ADAMTS genes responsible for Mendelian disorders.

Clinical symptoms and affected organs are extremely heterogeneous among hereditary diseases caused by ADAMTS genes. Phenotypic heterogeneity in these Mendelian disorders, despite shared enzymatic functions and similar functional defective features of germline mutations among ADAMTS family genes, is noticeable and it is alarming that paralogs within the ADAMTS gene family might function differently in different organs. Redundancy or compensation by other normal ADAMTS genes instead of affected ADAMTS genes is less likely to be anticipated based on the diverse symptoms among these disorders. Furthermore, subgrouping of ADAMTS genes according to shared unique protein domains did not fully correlate with the clinical presentations in affected patients, suggesting that organ-specific functions among individual ADAMTS genes are essential to their physiological roles.

To evaluate the possibility of yet-to-be-discovered Mendelian disorders caused by other ADAMTS genes, we applied two recently validated prediction algorithms [20,21] for missense burden estimation by regions across genes. Although only one algorithm predicted that four genes (*ADAMTS6, ADAMTS9, ADAMTS10, ADAMTS14*) possessed missense intolerance regions, pathogenic missense mutations in *ADAMTS9* and *ADAMTS10* indeed account for relatively high proportions among mutation types. Therefore, we suggest that *ADAMTS6* and *ADAMTS14*, which are not currently considered causative genes for any Mendelian disorders, might be responsible for novel hereditary disorders caused by pathogenic missense mutations. Furthermore, *ADAMTS6* and *ADAMTS2* also presented high pLI scores and low oe values for LoF using the gnomAD database. As nonsense and frameshift mutations in *ADAMTS2* are responsible for the Ehlers–Danlos syndrome, LoF mutations in *ADAMTS6* are expected to cause novel disease entities with diverse organotypic symptoms. Altogether, it will be interesting to examine whether any other ADAMTS proteins are involved in different forms of Mendelian diseases in the future.

When we applied the 2015 ACMG guideline [19] for pathogenicity interpretation in all reported ADAMTS family gene variants and compiled all published information on functional assays to evaluate the clinical validity of gene-disease relationships, the importance of an appropriate functional evaluation on diverse ADAMTS gene variants was once again confirmed in terms of accurately interpreting pathogenicity. Although *ADAMTS1* and *ADAMTS16* variants were suggested to be responsible for congenital mandibular prognathism and inherited hypertension, respectively, insufficient evidence for defective functions in ADAMTS proteins caused by variants and the scarcity of clinical reports did not allow strongly valid annotations for either the gene–disease relationship or variant pathogenicity. While the conventional genetic assessment of ADAMTS gene family variants should be considered in the context of prediction algorithm results, conservation status across species, and population allele frequencies, functional assays for clear defects appear to be the most important factor in the process of ACMG guideline application to establish a strong link between a pathogenic mutation and an associated Mendelian disorder.

## 5. Conclusions

In conclusion, we evaluated the genomic landscape and mutational spectrum of ADAMTS family genes in Mendelian disorders based on a gene evidence review of variants using publicly available databases and systematic literature reviews. Although eight ADAMTS family genes have a strong causal relationship with diverse Mendelian diseases in an autosomal recessive manner, there are additional possibilities for other ADAMTS family genes, such as *ADAMTS6*, to have a high potential in causing novel hereditary disorders based on our analysis. Despite an ultra-rare prevalence of pathogenic germline ADAMTS mutations responsible for genetic diseases (with the exception of *ADAMTS13*), it is important to accurately assess variants for their pathogenicity, together with metalloproteinase-specific functional assays for ADAMTS family proteins.

## Figures and Tables

**Figure 1 biomolecules-10-00449-f001:**
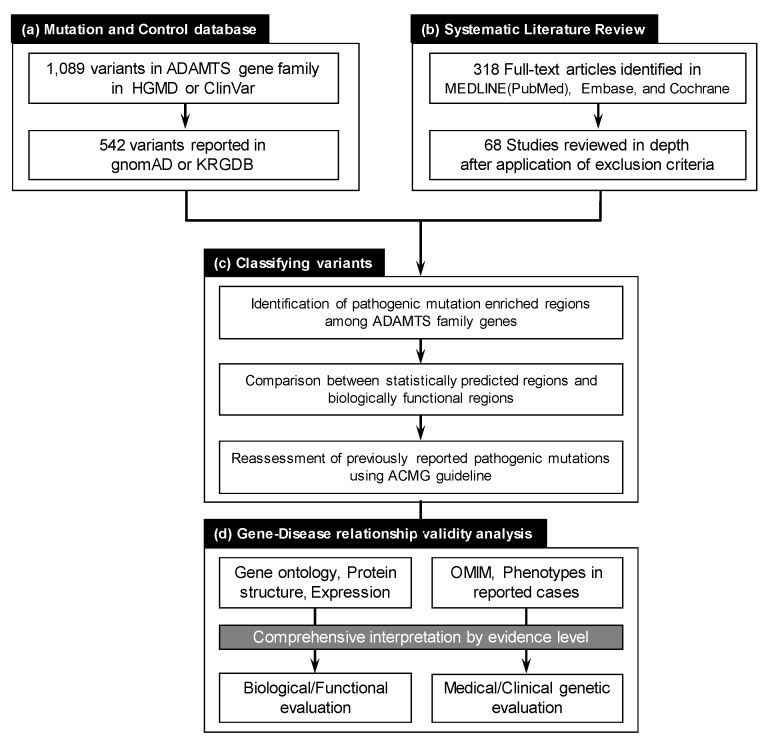
The overall workflow of the study. (**a**) ADAMTS family gene variants reported in public mutation and control databases were compiled. A total of 1089 variants were deposited in the Human Gene Mutation Database (HGMD) and ClinVar. Variants reported as “likely pathogenic” or “pathogenic” in ClinVar and as “DM” or “DM?” in HGMD were further examined. Of the 1089 variants, 541 were reported in gnomAD and KRGDB control datasets. (**b**) Systematic literature review in MEDLINE (PubMed), Embase, and the Cochrane Database of Systematic Reviews electronic databases revealed 318 full-text articles as of January 2020. Only 68 studies fulfilled our criteria and were fully reviewed in depth by two independent authors. (**c**) Variant classification, according to ACMG guidelines, along with mutation enriched region assessments using two different algorithms, were performed. (**d**) Evaluation of gene–disease relationships based on comprehensive evidence level interpretation were applied to all ADAMTS family genes.

**Figure 2 biomolecules-10-00449-f002:**
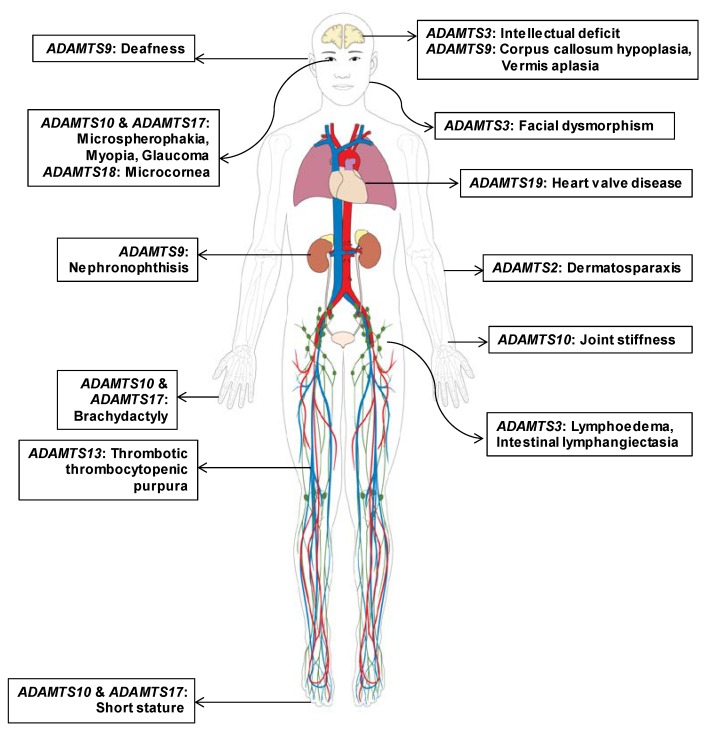
Clinical findings and distribution of affected organs in individuals with Mendelian disorders caused by pathogenic mutations in different ADAMTS family genes. Predominant and recurrently reported clinical presentations of eight ADAMTS genes with a strong causal relationship to Mendelian disorders are marked according to organ. Heterogeneous distribution of affected organ types indicates the phenotypic heterogeneity among hereditary disorders caused by pathogenic germline mutations in ADAMTS genes.

**Figure 3 biomolecules-10-00449-f003:**
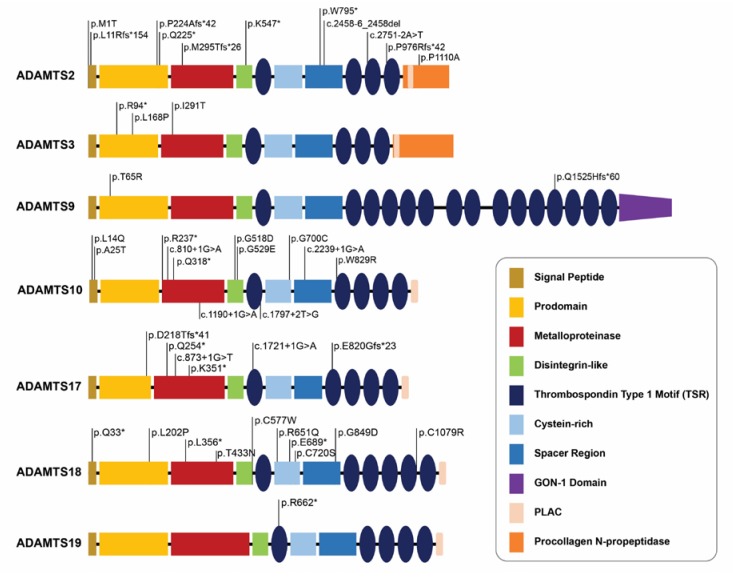
Comparative analysis of pathogenic mutation loci across ADAMTS genes with strong relationship with specific Mendelian disorders. Positional annotations of pathogenic mutations in ADAMTS genes by protein functional domains indicate the wide distribution of mutations and absence of hot spots. “*” (asterisk) indicates a translation termination codon.

**Table 1 biomolecules-10-00449-t001:** ADAMTS family genes responsible for Mendelian disorders with strong evidence as of 2020.

Gene	Disease	Inheritance Mode	Other Genes	Mutational Spectrum	Number of Reported Patients (Family)	Major References (PMID)
*ADAMTS2*	Ehlers–Danlos syndrome, dermatosparaxis type	Autosomal recessive	none	nonsense, frameshift	10 (10)	26765342
*ADAMTS3*	Hennekam syndrome	Autosomal recessive	*FAT4, CCBE1*	nonsense, missense	3 (2)	28985353
*ADAMTS9*	Nephronophthisis-related ciliopathy	Autosomal recessive	none	frameshift, missense	2 (2)	30609407
*ADAMTS10*	Weill–Marchesani syndrome	Autosomal recessive	*LTBP2, FBN1*	nonsense, missense, splice site	8 (7)	18567016, 25469541
*ADAMTS13*	Thrombotic thrombocytopenic purpura (Upshaw–Schulman syndrome)	Autosomal recessive	none	missense, nonsense, frameshift, splice site	more than 200	30770395, 30792199
*ADAMTS17*	Weill–Marchesani-like syndrome (Weill–Marchesani syndrome 4)	Autosomal recessive	none	nonsense, frameshift, splice site	4 (4)	19836009
*ADAMTS18*	Microcornea, myopic chorioretinal atrophy and telecanthus	Autosomal recessive	none	missense, nonsense	7 (5)	23356391, 23818446
*ADAMTS19*	Nonsyndromic heart valve disease	Autosomal recessive	none	nonsense, exonic deletion	4 (2)	31844321

**Table 2 biomolecules-10-00449-t002:** Evidence-based evaluation of the clinical validity of ADAMTS family genes in gene–disease relationships.

Gene	Gene Ontology (Shared Biological Processes)	Gene Ontology (Shared Molecular Functions) *	Protein Domain (Major Functional Domain, ADAMTS Backbone Domain Shared)	Expression Database (Uniprot, Human Protein Atlas)	Functional Assay Availability
*ADAMTS1*	Integrin-mediated signaling pathway Negative regulation of cell population Proliferation	Heparin binding Zinc ion binding	TSP type-1 repeats 2	Ovary, Immune cells, Facial skeletal	Extracellular region or secreted
*ADAMTS2*	Collagen catabolic process Collagen fibril organization Protein processing	Zinc ion binding	TSP type-1 repeats 3, Procollagen amino propeptidases	Connective tissue	NA
*ADAMTS3*	Collagen catabolic process Collagen fibril organization Protein processing	Heparin binding Zinc ion binding	TSP type-1 repeats 3, Procollagen amino propeptidases	Extremities	Extracellular region or secreted, Immunoprecipitation
*ADAMTS4*	Extracellular matrix Disassembly Proteolysis	Metal ion binding Metallopeptidase activity	None	Adipose tissue, CNS	NA
*ADAMTS5*	Extracellular matrix Disassembly Proteolysis	Extracellular matrix binding Heparin binding Integrin binding Metallopeptidase activity Zinc ion binding	TSP type-1 repeats 1	Breast, Placenta, Heart	NA
*ADAMTS6*	NA	Metal ion binding Metallopeptidase activity	TSP type-1 repeats 4, PLAC	Placenta, Brain, Ovarian follicle cells	NA
*ADAMTS7*	Cellular response to BMP stimulus Cellular response to tumor necrosis factor Negative regulation of chondrocyte Differentiation Proteolysis involved in cellular protein catabolic process	Metal ion binding Metallopeptidase activity	TSP type-1 repeats 7, Mucin-proteoglycans, PLAC	Heart muscle	NA
*ADAMTS8*	Negative regulation of cell population proliferation	Heparin binding Integrin binding Metallopeptidase activity Zinc ion binding	TSP type-1 repeats 1	Gallbladder, Lung	NA
*ADAMTS9*	Endothelial cell–matrix adhesion Extracellular matrix organization Positive regulation of melanocyte Differentiation Proteolysis	Metallopeptidase activity Zinc ion binding	TSP type-1 repeats 14, GON-1	Kidney, Adipose tissue	Extracellular region or secreted
*ADAMTS10*	NA	Metal ion binding	TSP type-1 repeats 4, PLAC	Connective tissue, Skin	Extracellular region or secreted, N-linked deglycosylation assay
*ADAMTS12*	Cell–matrix adhesion Cellular response to BMP stimulus Cellular response to tumor necrosis factor Negative regulation of chondrocyte differentiation Proteolysis involved in cellular protein catabolic process	Metal ion binding	TSP type-1 repeats 7, Mucin-proteoglycans, PLAC	NA	NA
*ADAMTS13*	Cell-matrix adhesion Cellular response to tumor necrosis factor Integrin-mediated signaling pathway Protein processing Proteolysis	Integrin binding Metallopeptidase activity Zinc ion binding	TSP type-1 repeats 7, CUB domain	Liver, Blood	Extracellular region or secreted, Beta-galactosidase activity
*ADAMTS14*	Collagen catabolic process Collagen fibril organization	Metal ion binding	TSP type-1 repeats 3, Procollagen amino propeptidases	Brain, Gallbladder, Placenta	NA
*ADAMTS15*	NA	Extracellular matrix binding Heparin binding Zinc ion binding	TSP type-1 repeats 2	Adipose tissue, Luminal membranes in the gastrointestinal tract	NA
*ADAMTS16*	NA	Metal ion binding	TSP type-1 repeats 5, PLAC	Kidney, Brain, Ovary	NA
*ADAMTS17*	NA	Metal ion binding	TSP type-1 repeats 4, PLAC	Lymphoid tissue, Connective tissue	Extracellular region or secreted
*ADAMTS18*	NA	Metal ion binding	TSP type-1 repeats 5, PLAC	Eye, Adipose tissue, Brain, Placenta, Extravillous trophoblasts, CNS, Bone	Medaka fish model (ocular), In-situ hybridization
*ADAMTS19*	NA	Metal ion binding	TSP type-1 repeats 4, PLAC	Cervix, Uterine, Endometrium, Smooth muscle, Ovary	NA
*ADAMTS20*	Extracellular matrix organization Positive regulation of melanocyte differentiation	Zinc ion binding	TSP type-1 repeats 14, GON-1	Brain, Placenta, Retina, Testis	NA

* All ADAMTS family proteins present molecular metalloendopeptidase activity function. NA, not available.

**Table 3 biomolecules-10-00449-t003:** Mutational spectrum of ADAMTS family genes in Mendelian hereditary disorders in current databases and literature.

Gene	Disease Database						Prediction Algorithms		Population Database		
	Number of SNV Mutations			Number of CNV Mutations			PER-Viewer	MTR-Viewer	gnomAD		
	HGMD *	ClinVar **	Total ***	HGMD *	ClinVar **	Total ***	Pathogenic Mutation Enriched Region	Missense Intolerance Regions	oe value_ Missense	oe value_ LoF	pLI Score
*ADAMTS1*	4	0	4	0	0	0	no (family1)	no	0.92	0.2	0.72
*ADAMTS2*	9	8	12	3	0	3	no (family1)	no	0.85	**0.18**	**0.97**
*ADAMTS3*	4	3	4	0	0	0	no (family1)	no	0.94	0.44	0
*ADAMTS4*	0	0	0	0	0	0	no (family1)	no	0.86	0.46	0
*ADAMTS5*	1	0	1	0	0	0	no (family1)	no	0.91	0.46	0
*ADAMTS6*	4	0	4	0	0	0	no (family1)	Yes	0.65	**0.12**	**1**
*ADAMTS7*	1	0	1	0	0	0	no (family2)	no	0.94	0.49	0
*ADAMTS8*	0	0	0	0	0	0	no (family1)	no	0.93	0.63	0
*ADAMTS9*	3	0	3	0	0	0	no (family3)	Yes	0.92	0.29	0
*ADAMTS10*	14	7	14	0	0	0	no (family1)	Yes	0.62	0.2	0.84
*ADAMTS12*	0	0	0	0	0	0	no (family2)	no	0.91	0.53	0
*ADAMTS13*	179	38	216	9	0	9	no (family4)	no	0.85	0.52	0
*ADAMTS14*	0	0	0	0	0	0	no (family1)	Yes	0.97	0.53	0
*ADAMTS15*	1	0	1	0	0	0	no (family1)	no	0.94	0.47	0
*ADAMTS16*	2	0	2	0	0	0	no (family1)	no	0.96	0.29	0
*ADAMTS17*	7	5	7	1	3	4	no (family1)	no	1.16	0.55	0
*ADAMTS18*	14	4	14	1	4	5	no (family1)	no	1.38	0.74	0
*ADAMTS19*	3	0	3	0	0	0	no (family1)	no	0.76	0.38	0
*ADAMTS20*	0	0	0	0	0	0	no (family3)	no	1.06	0.69	0

* HGMD: small nucleotide variants annotated as “DM?” and “DM”. ** ClinVar: small nucleotide variants annotated as “likely pathogenic” and “pathogenic”. *** Excluding the overlapping variants. Bolded numbers indicate the satisfaction of suggested criteria for loss of function mechanism.

**Table 4 biomolecules-10-00449-t004:** Reassessment of previously reported pathogenic mutations in ADAMTS family genes using the American College of Medical Genetics and Genomics (ACMG) variant interpretation guideline.

Gene	Tran- script	Nucleotide Change	Amino Acid Change	Conservation *				Population Database			Prediction Algorithms		ACMG Guideline	
				Mm	Gg	Xt	Dr	gnomAD_all	gnomAD_ maxP	db SNP	SIFT	PP2	Final Class	Component
*ADAMTS2*	NM_014244.4	c.2T>C	p.M1T	M	E	_	_	none		none			Pathogenic	PVS1, PM2, PM3
*ADAMTS2*	NM_014244.4	c.673C>T	p.Q225*	na	na	na	na	0.0150%	ASJ:0.30%	rs137853146			Pathogenic	PVS1, PM1, PP4
*ADAMTS2*	NM_014244.4	c.2384G>A	p.W795*	na	na	na	na	none		rs137853147			Pathogenic	PVS1, PM2, PP4
*ADAMTS2*	NM_014244.4	c.3328C>G	p.P1110A	P	L	_	_	none		none	Tol (0.29)	Ben (0.001)	Likely pathogenic	PM2, PP1, PP2, PP4, PP5
*ADAMTS2*	NM_014244.4	c.2751-2A>T	NA	na	na	na	na	none		none			Pathogenic	PVS1, PM2, PP4
*ADAMTS2*	NM_014244.4	c.884_887del	p.M295Tfs*26	na	na	na	na	none		none			Pathogenic	PVS1, PM2, PP4
*ADAMTS2*	NM_014244.4	c.2458-6_2458del	NA	na	na	na	na	none		rs1057517277			Pathogenic	PVS1, PM2, PP1, PP4
*ADAMTS2*	NM_014244.4	c.2927_2928del	p.P976Rfs*42	na	na	na	na	none		none			Pathogenic	PVS1, PM2, PP4
*ADAMTS2*	NM_014244.4	c.669dup	p.P224Afs*42	na	na	na	na	none		rs748037345			Pathogenic	PVS1, PM2, PP4
*ADAMTS2*	NM_014244.4	c.1638dup	p.K547*	na	na	na	na	none		none			Pathogenic	PVS1, PM2, PP4
*ADAMTS2*	NM_014244.4	c.32del	p.L11Rfs*154	na	na	na	na	none		none			Pathogenic	PVS1, PM2, PP4
*ADAMTS3*	NM_014243.2	c.280C>T	p.R94*	na	na	na	na	0.0004%	AFR:0.0062%	rs747975445			Pathogenic	PVS1, PP1, PP4
*ADAMTS3*	NM_014243.2	c.503T>C	p.L168P	L	L	L	L	0.0004%	NFE:0.00090%	rs1177851177	Del (0.01)	Dam (1.000)	Pathogenic	PM2, PM3, PP3, PP4
*ADAMTS3*	NM_014243.2	c.872T>C	p.I291T	I	I	I	I	none		rs61757480	Del (0.01)	Dam (1.000)	Pathogenic	PM2, PM3, PP3, PP4
*ADAMTS9*	NM_182920.1	c.194C>G	p.T65R	T	T	S	T	0.0240%	ASJ:0.096%	rs192420947	Del (0.01)	Dam (0.559)	Likely pathogenic	PS3, PP1, PP3
*ADAMTS9*	NM_182920.1	c.4575_4576del	p.Q1525Hfs*60	na	na	na	na	none		none			Pathogenic	PVS1, PS3, PM2
*ADAMTS10*	NM_030957.3	c.41T>A	p.L14Q	L	_	_	_	none		none	Del (0.01)	Ben (0.090)	Pathogenic	PS3, PM2, PP1, PP2, PP4, PP5
*ADAMTS10*	NM_030957.3	c.73G>A	p.A25T	A	_	_	_	0.0032%	SAS:0.0098%	rs121434358	Tol (0.05)	Ben (0.058)	Pathogenic	PS3, PM3, PP1, PP2, PP4, PP5
*ADAMTS10*	NM_030957.3	c.709C>T	p.R237*	na	na	na	na	0.0004%	EAS:0.0054%	rs121434357			Pathogenic	PVS1, PM3, PP4
*ADAMTS10*	NM_030957.3	c.952C>T	p.Q318*	na	na	na	na	none		rs121434359			Pathogenic	PVS1, PM2, PP4
*ADAMTS10*	NM_030957.3	c.1553G>A	p.G518D	G	G	G	G	none		rs267606636	Del (0)	Dam (1.000)	Likely pathogenic	PM2, PP1, PP3, PP4, PP5
*ADAMTS10*	NM_030957.3	c.1586G>A	p.G529E	G	G	G	G	0.0004%	NFE:0.00090%	none	Del (0)	Ben (0.270)	Likely pathogenic	PM2, PP1, PP4, PP5
*ADAMTS10*	NM_030957.3	c.2098G>T	p.G700C	G	G	G	G	none		rs267606637	Del (0)	Dam (1.000)	Likely pathogenic	PM2, PP1, PP3, PP4, PP5
*ADAMTS10*	NM_030957.3	c.2485T>A	p.W829R	W	W	W	W	none		none	Del (0)	Dam (0.999)	Likely pathogenic	PM2, PP1, PP3, PP4, PP5
*ADAMTS10*	NM_030957.3	c.810+1G>A	NA	na	na	na	na	0.0007%	ASJ:0.0097%	rs387906266			Pathogenic	PVS1, PP1, PP4
*ADAMTS10*	NM_030957.3	c.1190+1G>A	NA	na	na	na	na	none		rs431825170			Pathogenic	PVS1, PM2, PP4
*ADAMTS10*	NM_030957.3	c.1797+2T>G	NA	na	na	na	na	none		none			Pathogenic	PVS1, PM2, PP4
*ADAMTS10*	NM_030957.3	c.2239+1G>A	NA	na	na	na	na	0.0004%	AFR:0.0062%	rs782338897			Pathogenic	PVS1, PP1, PP4
*ADAMTS17*	NM_139057.3	c.760C>T	p.Q254*	na	na	na	na	none		rs267606638			Pathogenic	PVS1, PM2, PP4
*ADAMTS17*	NM_139057.3	c.1051A>T	p.K351*	na	na	na	na	none		none			Pathogenic	PVS1, PM2, PP4
*ADAMTS17*	NM_139057.3	c.873+1G>T	NA	na	na	na	na	none		none			Pathogenic	PVS1, PM2, PP4
*ADAMTS17*	NM_139057.3	c.1721+1G>A	NA	na	na	na	na	0.0032%	NFE:0.0071%	rs749116256			Pathogenic	PVS1, PP1, PP4
*ADAMTS17*	NM_139057.3	c.652delG	p.D218Tfs*41	na	na	na	na	none		none			Pathogenic	PVS1, PM2, PP4
*ADAMTS17*	NM_139057.3	c.2458dupG	p.E820Gfs*23	na	na	na	na	none		rs387906291			Pathogenic	PVS1, PM2, PP4
*ADAMTS18*	NM_199355.3	c.97C>T	p.Q33*	na	na	na	na	none		rs397515469			Pathogenic	PVS1, PM2, PP4
*ADAMTS18*	NM_199355.3	c.605T>C	p.L202P	L	L	L	I	none		rs397515468	Del (0.01)	Dam (0.992)	Likely pathogenic	PM2, PP1, PP3, PP4, PP5
*ADAMTS18*	NM_199355.3	c.1067T>A	p.L356*	na	na	na	na	none		none			Pathogenic	PVS1, PM2, PP4
*ADAMTS18*	NM_199355.3	c.1298C>A	p.T433N	T	T	T	T	none		none	Del (0.02)	Dam (1.000)	Likely pathogenic	PM2, PP1, PP3, PP4, PP5
*ADAMTS18*	NM_199355.3	c.1731C>G	p.C577W	C	C	C	C	none		rs148319220	Del (0)	Dam (1.000)	Likely pathogenic	PM2, PP1, PP3, PP4, PP5
*ADAMTS18*	NM_199355.3	c.1952G>A	p.R651Q	R	R	R	R	none		rs866074735	Del (0)	Dam (0.921)	Likely pathogenic	PM2, PP1, PP3, PP4, PP5
*ADAMTS18*	NM_199355.3	c.2065G>T	p.E689*	na	na	na	na	none		rs397515467			Pathogenic	PVS1, PM2, PP4
*ADAMTS18*	NM_199355.3	c.2159G>C	p.C720S	C	C	C	C	0.0004%	SAS:0.0033%	rs749517658	Del (0)	Dam (1.000)	Likely pathogenic	PM2, PP1, PP3, PP4, PP5
*ADAMTS18*	NM_199355.3	c.2546G>A	p.G849D	G	G	_	G	0.0004%	EAS:0.0054%	rs1417470741	Tol (0.44)	Dam (0.838)	Likely pathogenic	PM2, PP1, PP3, PP4, PP5
*ADAMTS18*	NM_199355.3	c.3235T>C	p.C1079R	C	C	C	C	0.0004%	NFE:0.00090%	rs1268581022	Del (0)	Dam (0.999)	Likely pathogenic	PM2, PP1, PP3, PP4, PP5
*ADAMTS19*	NM_133638.4	c.1984C>T	p.R662*	na	na	na	na	0.0008%	NFE:0.0018%	rs772148624			Pathogenic	PVS1, PS3

Abbreviations: Mm, *Mus musculus*; Gg, *Gallus gallus*; Xt, *Xenopus tropicalis*; Dr, *Danio rerio*; NA, not available; gnomAD_maxP, maximum minor allele frequency among sub-ethnic populations in the gnomAD database; ASJ, Ashkenazi Jewish; AFR, African; NFE, non-Finnish European; SAS, South Asian; EAS, East Asian; SIFT, Sorting Intolerant From Tolerant; PP2, PolyPhen-2; Tol, tolerated; Del, deleterious; Ben, benign; Dam, damaging; ACMG, American College of Medical Genetics and Genomics; PVS, pathogenic very strong; PS, pathogenic strong; PM, pathogenic moderate; PP, pathogenic supporting. * Codes in conservation columns represent the corresponding amino acids in four species at the positions of mutation according to standard amino acid abbreviations by IUPAC–IUB Joint Commission on Biochemical Nomenclature.

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
