# Peer review of "Genomic Landscape and Mutational Spectrum of ADAMTS Family Genes in Mendelian Disorders Based on Gene Evidence Review for Variant Interpretation"

_biomolecules, 2020, doi:10.3390/biom10030449_

Round 1

Reviewer 1 Report

I congratulate the authors for this excellent work. This is an important effort towards understanding the landscape of ADAMTS family gene variants.

The authors evaluated all reported ADAMTS family gene variants in ClinVar and HGMD mutation databases, as well as in recent literature on Mendelian hereditary disorders. They reassessed 299 variants using the ACMG 2015 guidelines. This review provides also evidences to associate each gene with a specific disorder.

The study contains a lot of data and information that can be useful in a daily practice of genotyping interpretations especially since this family is still poorly known. The manuscript is very written so easy to read and to follow.

Minor comments:

-Line 64 : rephrase please, this sentence is not clear to me. “were selected as HGMD professional curators that presumably categorized variant pathogenicity “

-For all databases used, please, add the website address (Mutalyzer Name Checker, HGMD…) and references when available.

-Please, detail what are oe values and pLI scores: observed/expected; probability of being loss-of-function intolerant…same, it’s not known for a lay audience what are PER viewer and MTR viewer? add a sentence to explain their utility

-Line 252: what is TPP acronym?

-Just a suggestion: Fig S1 could be a main figure

-Line 208, correct ADMATS10

Reviewer 2 Report

The authors provide a structural and functional characterization of the ADAMTS family genes, especially regarding their association with Mendelian hereditary disorders, based on an extensive revision of genetic variant evidence. They gather information from the ClinVar and HGMD mutation databases, as well as the literature.

The manuscript is nicely written and the information provided is well supported and useful for scientists interested in this gene family.

Major comments:

  • The authors provide good explanations on how they select the variants from ClinVar and HGMD (lines 63-66; Fig 1a & b), but they do not provide explicit details on how the classify the variants and how they perform the validity analysis (Fig 1c & d). I feel like these need more detailed explanations.
  • “this algorithm suggested that there were no definitive pathogenic mutation enriched regions for missense mutations in any ADAMTS gene” – needs specification of statistical test significance (lines 205-206)
  • “in contrast, ADAMTS6, ADAMTS9, ADAMTS10, and ADAMTS14 appeared to have missense mutation enriched regions according to MTR-viewer” – needs more details rather just qualitative observation (lines 206-208)

Minor comments:

  • I found tools/databases that the authors use and for which they do not provide proper citations: Mutalyzer Name Checker (line 67), MEDLINE (line 82), Embase (line 83), Cochrane Database of Systematic Reviews (line 83), OMIM (line 94), HPO (line 96), GO (line 97), DAVID (line 98), Ensembl (line 99), UniProt (line 100), Human Protein Atlas (line 100)
  • Typos: MEDLINE (line 76), missing N
  • Table 4 and Supplementary Table 1: The 4 columns corresponding to “Conservation” have codes that are not properly described in the abbreviations section.
